# A New Fractional-Order Constitutive Model and Rough Design Method for Fluid-Type Inerters

**DOI:** 10.3390/ma18112556

**Published:** 2025-05-29

**Authors:** Yandong Chen, Ning Chen

**Affiliations:** 1College of Intelligent Equipment Engineering, Wuxi Taihu University, Wuxi 214064, China; 2College of Mechanical and Electronic Engineering, Nanjing Forestry University, Nanjing 210037, China

**Keywords:** fluid-type inerter, fractional-order inerter, fractional-order model, multiphase mechanical characteristics

## Abstract

The understanding and application of fluid-type inerters by scholars have been on the rise. However, due to their intricate multiphase mechanical properties, existing models still have considerable room for improvement. This study presents two fractional-order models and conducts parameter identification by integrating them with classical experimental data. The first model is an independent fractional-order model. In comparison with traditional models, it demonstrates significantly higher fitting accuracy in frequency regions beyond the ultra-low frequency range. The second model is a segmented fractional-order model, which determines segments according to critical frequencies. Although this model enhances the overall fitting accuracy, it also leads to increased complexity. To tackle this complexity issue, a rough design strategy is proposed to minimize the critical frequency. Research indicates that under such a strategy, the inertial effect dominates the behavior of the fluid inerter. Even when the independent fractional-order model is used, a high fitting accuracy can be achieved. Consequently, by designing the structural parameters and fluid medium of the fluid inerter based on the rough design strategy, the model can be simplified. Moreover, compared with traditional nonlinear inerter models, the transfer function and eigenvalue analysis methods can be effectively applied. This enables the acquisition of more comprehensive theoretical research results, thereby greatly facilitating theoretical analysis.

## 1. Introduction

The term “Inerter” was first proposed by Professor Smith in 2002 [1]. Unlike traditional mass models, the output force of an inerter is related to its relative acceleration at its two endpoints. The ratio of the force it represents to the relative acceleration between its two endpoints is a constant, and the constant ratio is named “Inertance”. Inerters are mainly divided into two categories: mechanical inerters [1,2] and fluid inerters. Smith et al. [3] invented a new type of fluid inerter with a spiral tube as the main accessory. Compared with mechanical inerters, it has the advantages of simple structure, small volume, large inertia capacity coefficient, no backlash and eccentricity problems, and easy adaptation to various passive network layouts. It is these advantages that have prompted scholars to propose various hydraulic inerters based on them. Shen et al. [4,5] proposed a novel electromechanical inerter composed of a hydraulic piston inertial device and a linear motor. Zhang et al. [6] proposed that a diamond-shaped structure inerter can be modeled as a mem-inerter, and its inertial coefficient changes with the relative displacement of the two ends of the inerter. Yang et al. [7] established a theoretical model of a suspension system based on a liquid-gas inerter and developed a prototype of the liquid-gas inerter. Comparative bench tests were conducted to verify the accuracy of the simulation.

The models of fluid-type inerters are mostly based on empirical formulas obtained from different classical experiments and combined for fitting. To improve the fitting accuracy of fluid inerter models containing spiral tubes, scholars have proposed various experimental devices and identification methods. Liu et al. [8] proposed a new method for accurately measuring the parasitic damping of spiral tube fluid inertial vessels using dual pressure gauges at the inlet and outlet of spiral tubes. The theoretical formula for nonlinear spiral tube damping was verified, and the consistency with experimental data was much better than previous studies. Subsequently, Liu et al. [9] proposed the structure of an external-spiral-tube-type hydraulic inerter. Based on the friction, flow pressure loss, piston inertia, and fluid inertia of the hydraulic inerter, a mathematical model of the hydraulic inerter was established, and the damping, inertia, and stiffness effects were directly mapped to their respective hydraulic counterparts. The former also separated the identification of friction, stiffness, and various damping effects according to the experimental sequence. De Domenico et al. [10,11] used experimental data to characterize the dynamics of fluid inerters, provided a simplified model suitable for practical seismic engineering applications, and verified the simplified model. The model only retained one inertia term and one nonlinear damper representing the power-law damping term.

Based on the literature above, it is found that most of the fluid inerter models with spiral tubes are fitted by the superposition of empirical formulas from classical experiments, and all of them have nonlinear terms. Although the comparison between simulation models and experimental data is generally effective, the literature [4,12] also points out that there are still some differences in the degree of agreement between low and high frequencies. Therefore, it may seem that each item corresponds to a specific counterpart in the container, but in reality, the fluid flow inside is complex, and whether it is laminar, turbulent, or turbulent, fractal characteristics will appear locally [13]. From the perspective of modeling fractional differential equations [14], simply using multiple different integer-order derivatives to describe multiphase mechanical properties is unsatisfactory. Westerlund [15] suggests using a unified fractional-order model when describing multiphase mechanical properties. The author of this article first proposed a fractional-order model to describe fluid inerters, and the results showed that the fractional-order model can simultaneously describe the multiphase characteristics of inertia and damping [14]. The use of fractional-order models to describe fluid inerters can not only solve the theoretical research difficulties of nonlinear and unfavorable fluid inerters but also broaden their application prospects [16,17,18].

Therefore, this article will verify the feasibility of using fractional-order models to describe fluid inerters through mature experimental data. The main chapter content is as follows: Section 2 is to prepare knowledge, namely introducing the definition and properties of fractional calculus and existing fluid-type inerter models; Section 3 proposes two fractional-order models and performs fitting and validation; and Section 4 is the conclusion of the entire text.

## 2. Preparation Knowledge

### 2.1. Definition and Properties of Fractional Calculus

In the past few decades, three primary definitions have dominated the theoretical framework of fractional calculus: the Riemann–Liouville (R-L) definition, the Caputo definition, and the Grünwald–Letnikov (G-L) definition. These formulas have mathematical equivalence under specific conditions. Riemann–Liouville (R-L) definition: Due to its concise analytical expressions and suitability for theoretical deduction, it dominates in pure mathematical research. However, its dependence on non-physical initial conditions (involving fractional derivatives of functions at the origin) limits its applicability in empirical research. Caputo definition: Distinguished by physically interpretable initial conditions (integer-order derivatives at the origin), it is consistent with classical mechanics principles. Its Laplace transform property simplifies the solving process in engineering and applied science, especially for initial value problems. This operational advantage enhances the practicality of simulating real-world phenomena. Grünwald–Letnikov (G-L) definition: By providing a specific time interpretation through backward differential approximation, it becomes the basis for numerical implementation. Although theoretically robust, its computational strength often makes it less popular in analytical environments compared to the R-L and Caputo formulas. Given the emphasis on physical interpretability and computational efficiency, the Caputo definition is used as the primary framework. Throughout the manuscript, D^α^ denotes the Caputo fractional differential operator (0 < α < 1), and D^−α^ represents the corresponding fractional integral operator. This choice ensures compatibility with the standard initial value problem formula while maintaining the tractability of the analysis. The specific definition of the Caputo derivative is as follows [19]:(1)Dαf(t)=1Γ(1−α)∫0tfn(τ)dτ(t−τ)α+1−n, α∈ℝ, n∈ℕ+, n − 1 < α < n
where Γ(·) is the Gamma function.

The following are common fractional calculus properties that will be used in this article [19].

(1)Set *λ*, *μ*∈ℝ, and 0 < *α*, *β* < 1, then we obtain(2)$$\left\{\begin{array}{l} D^{\alpha }\left[D^{\beta }x(t)\right]=D^{\beta }\left[D^{\alpha }x(t)\right]=D^{\alpha +\beta }x(t)\Rightarrow D^{\alpha +\beta -1}\left[\dot{x}(t)\right]\\ D^{\alpha }\left[\lambda x(t)+\mu g(t)\right]=\lambda D^{\alpha }x(t)+\mu D^{\alpha }g(t) \end{array}\right.$$
where the functions *x*(*t*) and *g*(*t*) and their derivatives are continuous in the interval $\left[t_{0},t\right]$.(2)Given the time scale *τ* = *ωt* and the function *x*(*t*) = *z*(*τ*), we obtain(3)$$\frac{D^{\alpha }x\left(t\right)}{Dt^{\alpha }}=\omega^{\alpha }\frac{D^{\alpha }z\left(\tau \right)}{D\tau^{\alpha }}$$(3)If *x*(*t*) is a trigonometric function and 0 < *α* < 1, then we obtain(4)$$\left\{\begin{array}{l} D^{\alpha }\left[a\sin\left(\omega t+\varphi \right)\right]=a\omega^{\alpha }\sin\left(\omega t+\varphi +\frac{\alpha \pi }{2}\right)\\ D^{\alpha }\left[a\cos\left(\omega t+\varphi \right)\right]=a\omega^{\alpha }\cos\left(\omega t+\varphi +\frac{\alpha \pi }{2}\right) \end{array}\right.$$

### 2.2. The Traditional Models of Fluid-Type Inerters

After conducting research on the output force models of mainstream fluid-type inerters containing spiral tube attachments, it was found that their commonality is that they all contain inertial force and damping force. The equivalent method of inertial force is basically the same, while the equivalent method of damping force for spiral tubes is different. There are two main models: the first model assumes that laminar flow is the main method to calculate the damping force caused by pressure difference inside the spiral tube [3,12], and the second model assumes that turbulence is the main factor in calculating the damping force caused by pressure difference inside a spiral tube [10,11]. Most experimental studies in the above literature indicate that Coulomb friction is close to a constant. At low frequencies, friction is the main force, and there is basically no inertial or resistive force, so the absolute error is very low. At high frequencies, the total output force is relatively large, the proportion of frictional force is small, and the relative error is not high, but the absolute error is greater than at low frequencies. Therefore, although the nonlinear model is generally effective, there is still a low degree of consistency within a certain frequency range. Reference [10] clearly states that it is uncertain whether the fluid motion inside the spiral tube is laminar, turbulent, or both, so further research and improvement are needed for its model.

To better understand the traditional model, schematic diagrams of the fluid inerter structures corresponding to the two references [10,12] are shown in Figure 1 below.

To compare and verify the fitting accuracy of the fractional-order model proposed in this article, typical integer-order inerter models (IOIMs) corresponding to the two hypotheses are first presented here. The first model based on the assumption of laminar flow is as follows [12]:(5)$$F=bx^{\prime \prime }+c_{1}{\left(x^{\prime }\right)}^{2}+c_{2}x^{\prime }+\mathrm{sign}\left(x^{\prime }\right)f_{0}$$
where $b=\frac{m_{\mathrm{hel}}}{1+\left(h/2\pi r_{4}\right)}{\left(\frac{S_{1}}{S_{2}}\right)}^{2},c_{1}=\frac{3\rho S_{1}^{3}}{4S_{2}^{2}},c_{2}=\frac{8\mu lS_{1}^{2}}{r_{3}^{2}S_{2}}$.

The second model based on turbulence assumption is as follows [10]:(6)$$F=bx^{\prime \prime }+c_{1}{\left(x^{\prime }\right)}^{1.75}+c_{2}{\left(x^{\prime }\right)}^{2}+c_{3}x^{\prime }+\mathrm{sign}\left(x^{\prime }\right)f_{0}$$
where $\begin{array}{l} b=\frac{m_{\mathrm{hel}}}{1+\left(h/2\pi r_{4}\right)}{\left(S_{1}/S_{2}\right)}^{2},\;c_{1}=\frac{0.0664\mu^{0.25}\rho^{0.75}lS_{1}}{r_{3}^{1.25}}{\left(S_{1}/S_{2}\right)}^{1.75},\;c_{2}=\frac{3\rho S_{1}^{3}}{4S_{2}^{2}},\;c_{3}=\frac{2\pi \mu r_{2}L}{\Delta r},\;\\ m_{\mathrm{hel}}\approx \rho lS_{2},\;S_{1}=\pi \left(r_{2}^{2}-r_{1}^{2}\right),\;S_{2}=\pi r_{3}^{2},\;l=n_{t}\sqrt{h^{2}+{\left(2\pi r_{4}\right)}^{2}},\;L=2n_{t}r_{3} \end{array}$.

In the two equations, *b* is the equivalent inertance, and the equivalent method is the same. In Equation (5), *c*_1_ and *c*_2_ are damping coefficients obtained by approximating laminar flow. In Equation (6), *c*_1_ and *c*_2_ are damping coefficients obtained by approximating turbulence. *c*_3_ is the thin-film shear friction force between the piston side wall and the surrounding cylinder liner, *f*_0_ is the amplitude of the Coulomb friction force between the piston and the cylinder wall, *S*_1_ is the effective area of the hydraulic cylinder, *S*_2_ is the section area of the helical channel, *l* is the unfolded length of the helical tube, *m*_hel_ is the equivalent mass of the fluid inside the helical tube, and *L* is the height of the helical tube component.

The other variables in the equivalent coefficient are the structural parameters of the inerter and the fluid parameters, as shown in Table 1.

## 3. Fractional-Order Inerter Model (FOIM)

### 3.1. The Proposal of an FOIM

When describing complex mechanical problems, compared to nonlinear models, fractional derivative models have clearer physical parameter meanings and more concise descriptions [20]. The empirical formulas used in fluid inerter models all have power-law functions (such as Equations (5) and (6)), but the mechanical constitutive relationship of the power-law function does not follow the standard “gradient” law. Its physical and mechanical evolution processes have obvious memory and path dependence properties, and fractional derivatives can better characterize these properties [14]. At present, relevant experiments on fluid inerters have shown that Coulomb friction is close to a constant, with a higher proportion at low frequencies and a lower proportion at high frequencies [21]. Therefore, this article proposes a fractional-order inerter model (FOIM) composed of fractional derivatives and Coulomb friction, in the specific form of(7)$$F=b\left(1+b_{0}\right)D^{\mu }x+\mathrm{sign}\left(x^{\prime }\right)f_{0}$$
where $D^{\mu }x$ is the fractional derivative term, *f*_0_ is the Coulomb friction force amplitude, *b* is the equivalent inertance defined by the traditional model, *b*_0_ is the correction coefficient, and *μ* is the derivative order (1 < *μ* < 2). Therefore, the parameters that need to be identified in this model are *b*_0_ and *μ*.

For general dynamical systems, when the derivative order is within the range of 0 < *α* < 1, it not only acts as a classical damping force but also as a restoring force [22]. Similarly, when the order of the derivative term is within the range of 1 < *μ* < 2, it acts as both a classical damping force and an inertial force [16]. According to the properties of fractional calculus, $D^{\mu }x$ is first transformed into $D^{\mu -1}\left[x^{\prime }\right]$, and then the integer equivalent method [23] is used to obtain the integer-order form of Equation (7) under harmonic displacement excitation (*x*(*t*) = *A*_0_sin (*ωt*)), as shown below:(8)$$F=b\left(1+b_{0}\right)\left(\omega^{\mu -2}\sin\frac{\mu -1}{2}\pi x^{\prime \prime }+\omega^{\mu -1}\cos\frac{\mu -1}{2}\pi x^{\prime }\right)+\mathrm{sign}\left(x^{\prime }\right)f,\;0<\mu -1<1$$

Compared with existing models, the actual equivalent coefficient in Equation (8) is not only related to the derivative order but also to the excitation frequency, with fewer identified parameters and clearer physical meanings. *b*(1 + *b*_0_) reflects the overall amplitude of inertia and damping, and *μ* reflects the proportion of inertia and damping [16].

### 3.2. Model Identification and Validation

This section combines known experimental data (extracted by the professional icon digitization tool GetData), identifies the parameters of the fractional-order model using a genetic algorithm [24], and compares them with existing models through simulation to verify the feasibility of the fractional-order model.

The model parameters *μ*, *b*_0_, and *f*_0_ identified by the genetic algorithm (GA) have a certain impact on the robustness and reliability of the model fitting due to their selection range. This article provides the following explanation on the selection of three parameters: (1) The setting of *μ* is related to frequency selection. At ultra-low frequencies, damping is clearly the main factor, and *μ* is set to 0–1.5, which is more appropriate. At other frequencies, inertia is the main factor, and *μ* is set between 1.5 and 2. (2) *b*_0_ refers to the addition of *b*_0_ times *b* to the traditional inertance *b*, with a range of 0–30. (3) *f*_0_ is a constant Coulomb friction force determined by low-frequency experiments. The Coulomb friction force amplitudes in references [10,12] are approximately 400 N and 550 N, respectively, and the equivalent inertance *b* is 370.38 and 73.82, respectively. The experimental data of the first and second groups in reference [12] were obtained under the action of sine displacement functions, with Ω = 0.5 Hz, *A*_0_ = 20 mm and Ω = 12 Hz, *A*_0_ = 5 mm, respectively; the third set of experimental data in reference [10] was obtained under the action of a sine displacement function with Ω = 3 Hz and *A*_0_ = 17.5 mm.

Based on the first two sets of experimental data, the corresponding fractional-order model parameters identified by the genetic optimization algorithm are *b*_0_ = 18.25, *μ* = 0.48 (this FIOIM is abbreviated as M1) and *b*_0_ = 0.57, *μ* = 1.89 (this FIOIM is abbreviated as M2), and based on this, the fractional-order model, integer-order model, and experimental curves were plotted, as shown in Figure 2.

Research has found that when the excitation frequency is 0.5 Hz, no matter how optimized the identification is, the derivative order is always less than one. Furthermore, using the model (M2) identified by 12 Hz to describe the low-frequency characteristics may result in significant errors. Therefore, M1 is suitable for describing the ultra-low frequency region, but as shown in Figure 2a, there are still some differences between the curves of traditional and fractional-order models and experimental data. From Figure 2b, the three curves in the figure have a high degree of overlap. The parameters identified at a frequency of 12 Hz are more in line with the requirements of the inerter, with an order close to two and a relatively small *b*_0_, indicating that both traditional and fractional-order models have high accuracy at this time.

To verify the effectiveness of M2 under other excitation frequencies and amplitudes, the comparison curves between M2 and existing models in reference [12] under excitation conditions of Ω at 3 Hz, 8 Hz, 15 Hz, and *A*_0_ at 10 mm, 5 mm, and 5 mm were provided, as shown in Figure 3.

Figure 3 shows that M2 has a certain phase difference compared to the model in reference [12] at low frequencies, and there is a small difference at the peak at other frequencies. Overall, it can achieve the fitting accuracy of traditional integer-order models, indicating that the fractional-order model has good applicability except for the ultra-low frequency region.

Based on the experimental data from the third group, the parameters of the fractional-order model identified using the same method as before are *b*_0_ = 0.087 and *μ* = 1.98. At this point, the derivative order approaches two, indicating a higher proportion of inertial effects and a lower proportion of damping effects. Similarly, the parameters were substituted into the fractional-order model and the existing model and compared with the experimental data, as shown in Figure 4. It can be seen from the figure that the fractional-order model can accurately fit the experimental data.

Similarly, to demonstrate the applicability of the fractional-order model, a comparison chart between the model and existing integer-order models is provided for other excitation frequencies (0.5 Hz, 8 Hz, and 12 Hz) and amplitudes (15 mm, 5 mm, and 5 mm), as shown in Figure 5. All three graphs show that the curves of the two models almost overlap, indicating that the fractional-order model fitting of the fluid inerter in reference [12], which mainly utilizes inertia, is more effective.

To compare the fitting accuracy of the model more accurately, the root mean square error (RMSE) is introduced as the evaluation index.(9)$$\mathrm{RMSE}=\sqrt{\sum_{i=1}^{m}{\left(y_{i}-{\bar{y}}_{i}\right)}^{2}/m}$$

In the formula, $y_{i}$ represents the data of the actual model, ${\bar{y}}_{i}$ represents the data of the comparative model, and *m* represents the total amount of data. To compare the proportion of RMSE in the average amplitude *A*_av_ of the comparison model, the following evaluation indicators are provided:(10)$$e_{d}=\frac{\mathrm{RMSE}}{A_{\mathrm{av}}}\times 100\%$$

Table 2 shows the *e*_d_ between the fractional-order model and experimental data in Figure 2 and Figure 4, as well as the *e*d between the integer-order model and experimental data. Table 3 shows the *e*d between the fractional-order model and the integer-order in Figure 3 and Figure 5.

From Table 2, the fitting accuracy of the fractional order model is close to or even exceeds that of the traditional models. Table 3 reflects the degree of closeness between the fractional-order model and the traditional model at other frequencies. Except for the first group, where the difference between the two models is large due to phase difference, the rest are relatively close, further indicating that the fractional-order model has certain applicability.

In addition, comparing the fractional-order model parameters of the two studies, the derivative order of the latter is 1.98, which is greater than the former’s 1.89. This conclusion is basically consistent with one of the research conclusions in references [3,25], that is, when the density is close, the flow with smaller kinematic viscosity (the latter, 0.0068, is less than the former, 0.027) has larger inertia and smaller damping, indicating a certain relationship between the parameters of the fractional-order model and the fluid parameters.

### 3.3. Segmented Fractional-Order Inerter Model (SFOIM)

From the previous analysis, the mechanical characteristics of ultra-low frequency and other frequency regions are different. To improve fitting accuracy, the following segmented fractional-order model is proposed:(11)$$F=\left\{\begin{array}{l} \Omega <\Omega_{\mathrm{low}}:b\left(1+b_{0}\right)D^{\alpha }x+\mathrm{sign}\left(x^{\prime }\right)f_{0},\alpha \in \left(0,1\right)\\ \Omega >\Omega_{\mathrm{low}}:b\left(1+b_{0}\right)D^{\mu }x+\mathrm{sign}\left(x^{\prime }\right)f_{0},\mu \in \left(1,2\right) \end{array}\right.$$
where Ω_low_ is the critical frequency.

Discrimination of critical frequency is the key to improving the accuracy of the model. Below, we will discuss the approximate discrimination method of critical frequency. According to the motion relationship of the fluid inerter, $A_{1}x^{\prime }=A_{2}u$, where $x^{\prime }$ is the piston velocity, *u* is the average velocity of the fluid in the spiral tube, and the approximate Reynolds number in the spiral tube can be calculated by the following equation [3]:(12)Re=2ρDhμu=Rvx′
where $R_{v}=\frac{2\rho D_{h}A_{1}}{\mu A_{2}}$.

In a straight pipe, the Reynolds number (Re) is used to determine whether the fluid flow is laminar or turbulent, and this transition occurs around Re = 2 × 103. As the experimental excitation is a harmonic displacement signal, *x* = *A*_0_ sin(*ωt* + *φ*_0_), and substituting this into the above equation can obtain the relationship between the Reynolds number amplitude *A*Re and the excitation frequency Ω.(13)Re=Rvx′=RvωA0cosωt+φ0⇒ARe=2πA0RvΩ

The fluid flowing along the curved pipeline is affected by centrifugal force, which forms a motion mode called secondary flow [3], as shown in Figure 6. This can be described to some extent by the Dean’s number (De) [26], which depends on the inertia and viscosity of the fluid, as well as the curvature ratio of the bending, 2*r*3/*r*4.(14)De=Re2r3/r4

Secondary flow is an example of a stable flow pattern, and it has been observed that as the Dean’s number increases, the critical Reynolds number of turbulence also increases. Therefore, the critical Reynolds number in a straight tube is directly used to identify whether secondary flow is generated in a helical tube has certain limitations. Rodman and Trenc [27] described rectangular cross-section spiral channels with different aspect ratios. When the Dean’s number range is 100 < De < 800, the flow channel is like an inertial fluid inerter. Reference [3] points out that based on their conclusion, it can be considered that the flow from 0 to 100 is laminar, with viscous friction being the main force. If it exceeds 800, other larger Dean numbers can be linearly extended, so De = 100 can be used as the critical point.

By referring to the relationship between the Reynolds number amplitude *A*_Re_ and the excitation frequency Ω, the relationship between the Dean number amplitude *A*_De_ and the excitation frequency Ω can be obtained:(15)$$A_{\mathrm{De}}=R_{v}\sqrt{2r_{3}/r_{4}}2\pi A_{0}\Omega$$

Based on the structural parameters from two references ([10] and [12]) and Equations (13) and (15), the variation curves of *A*_Re_ and *A*_De_ with Ω were plotted as shown in Figure 7. In the figure, the solid red line and the dotted purple line represent the curves for excitation amplitudes of 5 and 10 mm, respectively, according to reference [12]; the blue dashed line and orange dotted line represent the curves for excitation amplitudes of 5 and 10mm, respectively, according to reference [10].

From Figure 7, the critical frequency obtained using the dean number is smaller, indicating that turbulence is more likely to occur when using a helical tube. In addition, it can be observed that the excitation amplitude is larger, and the critical frequency is smaller. The critical frequency of reference [10] is much smaller than that of reference [12], as shown in Figure 5a, where the fitting accuracy is still high when Ω = 0.5 Hz.

Below are two sets of comparison graphs from reference [12] to verify the reliability of the approximate critical frequency obtained when the Dean’s number is equal to 100. Based on the identified parameters of two models (M1 and M2) in the previous section of reference [12], a comparative analysis chart was drawn for M1 and M2 with existing integer-order models when Ω < Ω_low_ (Figure 8) and Ω > Ω_low_ (Figure 9) were used. Figure 8 is obtained under displacement excitation of Figure 8a Ω = 0.3Hz, *A*_0_ = 5mm, and Figure 8b Ω = 0.5 Hz, *A*_0_ = 5 mm; Figure 9 is obtained under displacement excitation of Figure 9a Ω = 2 Hz, *A*_0_ = 5 mm, and Figure 9b Ω = 5 Hz, *A*_0_ = 5 mm.

From the comparison in Figure 7, the accuracy is relatively higher when using M1, while the comparison in Figure 9 shows that the accuracy is higher when using M2. For fluid inerters with structural parameters as described in reference [12], a segmented fractional-order model can be used, while reference [10], due to its particularly small critical frequency, can use the same fractional-order model. Therefore, approximating the critical frequency based on the critical dean’s number is feasible. By substituting the original variables into Equation (15), the relationship between frequency and parameters can be obtained as follows:(16)$$\Omega =\frac{1}{8\sqrt{2}\pi A_{0}}\frac{\mu }{\rho }\frac{r_{4}^{1/2}\left(r_{2}^{2}-r_{1}^{2}\right)}{r_{3}^{1/2}r_{3}^{3}}A_{\mathrm{De}}$$

In summary, although the segmented fractional-order model can have higher fitting accuracy than a single fractional-order model in the ultra-low frequency region, the critical frequency is small, and the error of ultra-low frequency is relatively small compared to friction. In practical research, the characteristics of ultra-low frequency are generally not discussed. Therefore, using an inertia-based fractional-order model can basically meet the requirements. In addition, although fractional-order models have fewer parameters, their fitting accuracy is close to or even exceeds that of existing models except for ultra-low frequency regions. Therefore, reasonably designing the structural parameters of fluid-type inerters to minimize the critical frequency can avoid segmented fractional-order models.

## 4. Conclusions

This article innovatively proposes the concept of critical frequency using classical theories and applies it to the design of fluid inerters with inertial characteristics as the focus. At the same time, fully utilizing the advantages of fractional-order models in fitting multiphase characteristics, two fractional-order models with clear physical meanings and more simplified parameters were constructed based on traditional fluid inerter models. The specific research results are as follows:(1)Research has shown that when using segmented fractional-order models for fluid inerters, the fitting accuracy in the ultra-low frequency region is better than that of independent fractional-order models. However, this high precision comes at the cost of increasing model complexity. This suggests that we need to balance the relationship between accuracy and model complexity in practical applications.(2)Research has found that when the critical frequency is small enough, the use of an independent fractional-order model for fluid inerters can meet practical engineering needs. Equation (16) can serve as a rough design principle for fluid inertial containers, providing a simple and effective reference for engineering design.

Overall, by designing the structural parameters of fluid inertia and fluid medium reasonably, the fractional-order model demonstrates good applicability. Compared with traditional nonlinear inertial models, fractional-order models can use transfer function and eigenvalue analysis methods to conduct theoretical research, which can obtain richer theoretical results and greatly facilitate theoretical analysis work. This not only deepens the theoretical understanding of fluid inertial containers but also provides strong support for the promotion and application of fluid inertial containers.

Overall, by designing the structural parameters and fluid medium of the fluid inerter reasonably, the fractional-order model demonstrates good applicability. Compared with traditional nonlinear inerter models, fractional-order models can use transfer function and eigenvalue analysis methods to conduct theoretical research, which can obtain richer theoretical results and greatly facilitate theoretical analysis work. This not only deepens the theoretical understanding of fluid inerters but also provides strong support for the promotion and application of fluid inerters.

## Figures and Tables

**Figure 1 materials-18-02556-f001:**
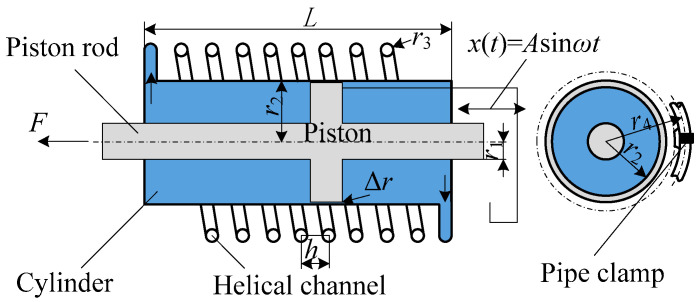
Schematic diagram of fluid inerter structure.

**Figure 2 materials-18-02556-f002:**
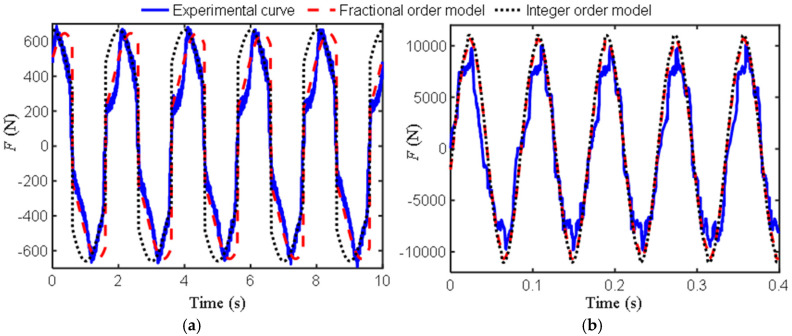
Curves of the output force of the inerter over time: (**a**) Ω = 0.5 Hz, *A*_0_ = 20 mm; (**b**) Ω = 12 Hz, *A*_0_ = 5 mm.

**Figure 3 materials-18-02556-f003:**
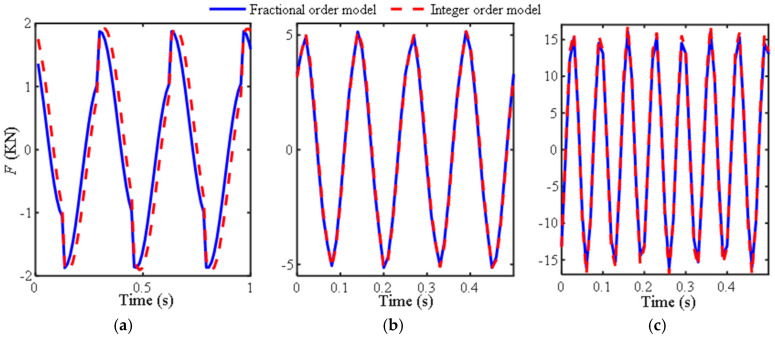
Curves of the output force of the inerter over time: (**a**) Ω = 3 Hz, *A*_0_ = 10 mm; (**b**) Ω = 8 Hz, *A*_0_ = 5 mm, (**c**) Ω = 15 Hz, *A*_0_ = 5 mm.

**Figure 4 materials-18-02556-f004:**
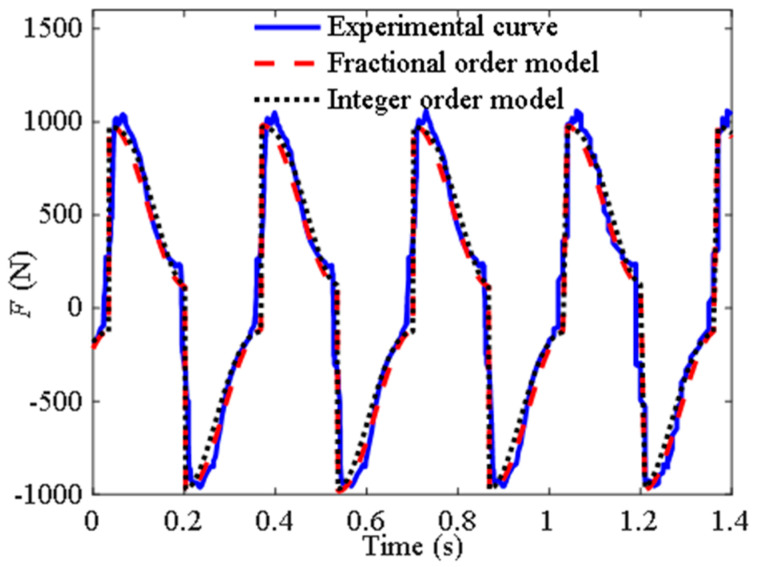
Curves of the output force of the inerter over time.

**Figure 5 materials-18-02556-f005:**
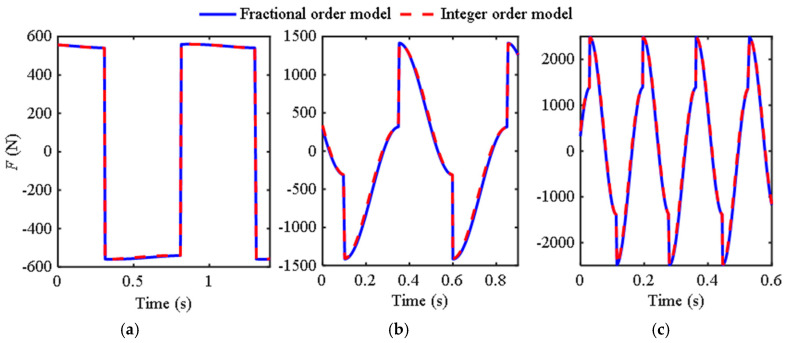
Curves of the output force of the inerter over time: (**a**) Ω = 0.5 Hz, *A*_0_ = 15 mm; (**b**) Ω = 8 Hz, *A*_0_ = 5 mm; (**c**) Ω = 12 Hz, *A*_0_ = 5 mm.

**Figure 6 materials-18-02556-f006:**
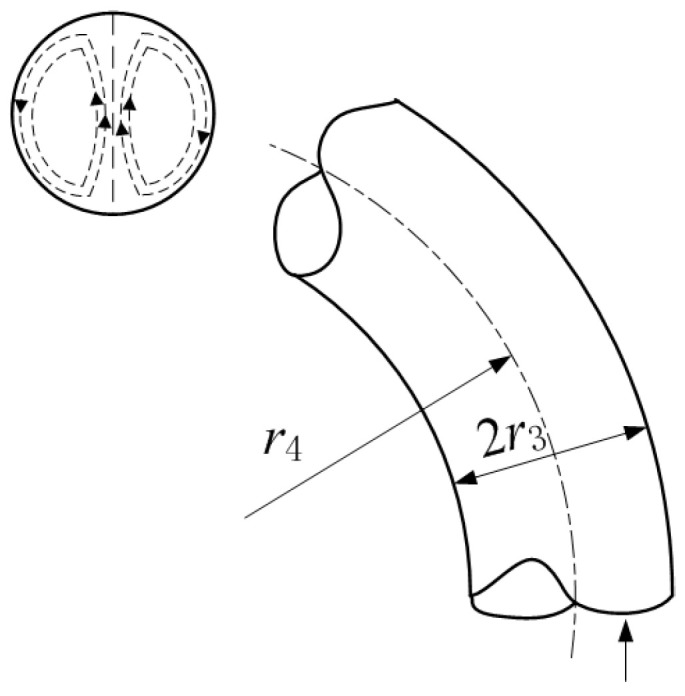
Secondary flow in a curved channel with a circular cross-section.

**Figure 7 materials-18-02556-f007:**
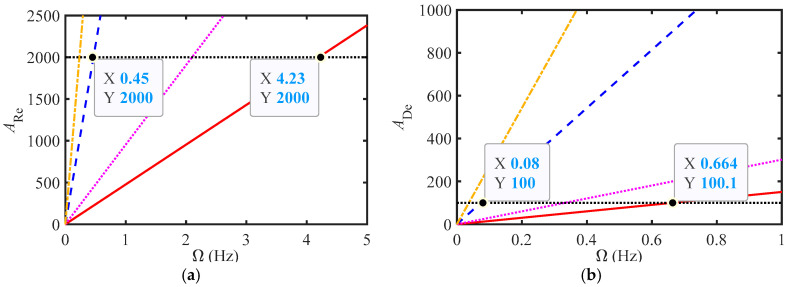
The variation curves of (**a**) *A*_Re_ and (**b**) *A*_De_ with Ω at different *A*_0_ values.

**Figure 8 materials-18-02556-f008:**
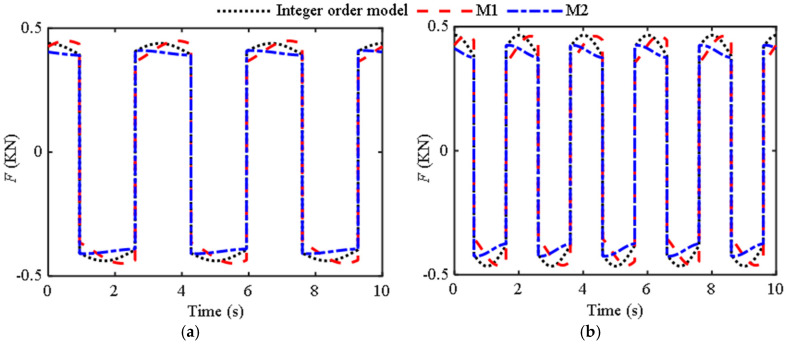
Comparison curves of different models when Ω < Ω_low_: (**a**) Ω = 0.3 Hz, *A*_0_ = 5 mm; (**b**) Ω = 0.5 Hz, *A*_0_ = 5 mm.

**Figure 9 materials-18-02556-f009:**
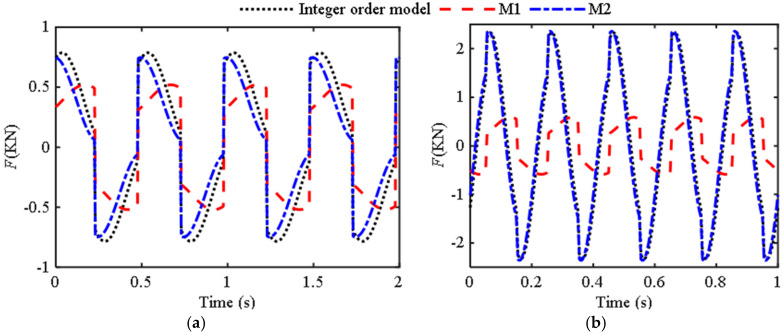
Comparison curves of different models when Ω > Ω_low_: (**a**) Ω = 2 Hz, *A*_0_ = 5 mm; (**b**) Ω = 5 Hz, *A*_0_ = 5 mm.

**Table 1 materials-18-02556-t001:** Parameters of inerters in references [10] and [12].

Value Name	Value [12]	Value [10]
Radius of the piston *r*_1_ (m)	0.012	0.014
Inner radius of the cylinder *r*_2_ (m)	0.028	0.025
Inner radius of the helical channel *r*_3_ (m)	0.005	0.006
Radius of the helix *r*_4_ (m)	0.1	0.12
Pitch of the helix *h* (m)	0.012	0.03
Clearance between the piston head and the cylinder wall Δ*r* (mm)	0	0.1
Circle number of helical channel *n*_t_	14	7
Oil density *ρ* (kg∙m^−3^)	800	802
Length of transition section *l*_0_ (m)	0.1	0
Viscosity of fluid *μ* (Pa∙s)	0.027	0.00168

**Table 2 materials-18-02556-t002:** The *e*_d_ between two models and experimental data in Figure 2 and Figure 4.

Ref. [12]: 0.5 Hz, 20 mm		Ref. [12]: 12 Hz, 5 mm		Ref. [10]: 3 Hz, 17.5 mm	
FOIM (M1)	IOIM	FOIM (M2)	IOIM	FOIM	IOIM
23.99%	31.95%	19.05%	21.2%	23.39%	23.44%

**Table 3 materials-18-02556-t003:** The *e*_d_ between the two models in Figure 3 and Figure 5.

Ref. [12]			Ref. [10]		
3 Hz, 10 mm	8 Hz, 5 mm	15 Hz, 5 mm	0.5 Hz, 15 mm	8 Hz, 5 mm	12 Hz, 5 mm
19.1%	2.6%	5.44%	0.2%	2.12%	1.85%

## Data Availability

The original contributions presented in the study are included in the article. Further inquiries can be directed to the corresponding authors.

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
