# Peer review of "A New Fractional-Order Constitutive Model and Rough Design Method for Fluid-Type Inerters"

_materials, 2025, doi:10.3390/ma18112556_

Round 1
Reviewer 1 Report
Comments and Suggestions for Authors
In this manuscript, the authors presented A new fractional order constitutive model and rough design method for fluid-type inerters. This study offers a significant contribution by offering two fractional-order models for fluid inerters and recognizing the complexity trade-off of the segmented approach. It also introduces a critical frequency based on the Dean number to increase accuracy in ultra-low-frequency regimes. The probabilistic design approach is very useful for engineering applications since it reduces the critical frequency and makes employing a more straightforward independent model possible.
Please review the entire manuscript to fix grammar and syntax errors, like in:
Line 64.- Turbulent is a duplicate in the paragraph.
Line 86.- derivation; The initial
Line 116.- low; At high
Line 120.- Reference [10] clearly states
Line 133.- the comma between the equations.
Line 184.- respectively; The third
The only concern is the data capture using DATAGRAPH. What is the precision or resolution employed in DATAGRAPH?
Overall, this valuable contribution to the field merits publication after addressing these points.
Author Response
Comments 1: Please review the entire manuscript to fix grammar and syntax errors, like in: Line 64.- Turbulent is a duplicate in the paragraph. Line 86.- derivation; The initial Line 116.- low; At high Line 120.- Reference [10] clearly states Line 133.- . Line 184.- respectively; The third |
Response 1: Thank you for pointing this out. We agree with this comment. Therefore, we have addressed the grammar errors in the text and have made improvements to all parts except for the ones mentioned above, marked in red font Such as lines 83-103, 124-131, 157, 199, etc in the new manuscript. For example:“In the past few decades, three primary definitions have dominated the theoretical framework of fractional calculus: Riemann Liouville (R-L) definition, Caputo definition, and Grünwald Letnikov (G-L) definition. These formulas have mathematical equivalence under specific conditions. Riemann Liouville (R-L) definition: due to its concise analytical expressions and suitability for theoretical deduction, it dominates in pure mathematical research. However, its dependence on non-physical initial conditions (involving fractional derivatives of functions at the origin) limits its applicability in empirical research. Caputo definition: distinguished by physically interpretable initial conditions (integer order derivatives at the origin), consistent with classical mechanics principles. Its Laplace transform property simplifies the solving process in engineering and applied science, especially for initial value problems. This operational advantage enhances the practicality of simulating real-world phenomena. Grünwald Letnikov (G-L) definition: by providing a specific time interpretation through backward differential approximation, it becomes the basis for numerical implementation. Although theoretically robust, its computational strength often makes it less popular in analytical environments compared to R-L and Caputo formulas. Given the emphasis on physical interpretability and computational efficiency, the Caputo definition is used as the primary framework. Throughout the manuscript: Dα denotes the Caputo fractional differential operator (0<α<1), D−α represents the corresponding fractional integral operator. This choice ensures compatibility with the standard initial value problem formula while maintaining the tractability of the analysis. The specific definition of Caputo derivative is as follows[19].” “Most experimental studies in the above literature indicate that Coulomb friction is close to a constant. At low frequencies, friction is the main force, and there is basically no inertial or resistive force, so the absolute error is very low. At high frequencies, the total output force is relatively large, the proportion of frictional force is small, and the relative error is not high, but the absolute error is greater than at low frequencies.”
|
Comments 2: The only concern is the data capture using DATAGRAPH. What is the precision or resolution employed in DATAGRAPH? |
Response 2: Thank you for pointing this out. We have provided the following explanation regarding the usage issue of DATAGRAPH. According to the smoothness of the curve, the total number of points taken is increased or decreased. As shown in Figure 2a, the curve is more complex at the mutation point and more points need to be added. In areas without jagged edges, fewer points can be taken, totaling 2500 points. In Figure 2b, there are fewer jagged edges, totaling 1600 points. Generally, automatic and manual modes are used to improve the accuracy of point taking, and overall, the accuracy is relatively high. |

Reviewer 2 Report
Comments and Suggestions for Authors
1. The abstract should first briefly describe the problem and its relevance. Otherwise, the reader will not understand at all what the article is about.
2. The classical mathematical model (equations 5-6) should be described in more detail. Not all notations are given. The physical meaning of each term in the expression for the force F should be explained in the diagram.
3. The work uses two expressions for the force, for laminar (5) and for turbulent flow (equation 6). How do the authors choose which one to choose?
4. The main problem with the work is that the use of a model with fractional dimension is not justified. It remains unclear to the end why the authors needed to use a model with fractional dimension. What physical meaning does it carry? The authors' assertion that it better describes the experiment is not convincing. In my opinion, this is just a question of approximation. We are considering a classical problem and it should be well described by the equations of classical mechanics. With the same success, using the least squares method, the parameters of the classical model can be fitted in the best possible way, and the classical model will describe the experiment better than the model with fractional dimension.
Author Response
Comments 1: The abstract should first briefly describe the problem and its relevance. Otherwise, the reader will not understand at all what the article is about.
|
Response 1: Thank you for pointing this out. We agree with this comment. Therefore, we have added a description of the role and existing issues of fluid type inerters, as shown in red font below: “The understanding and application of fluid type inerters by scholars have been on the rise. However, due to their intricate multiphase mechanical properties, existing models still have considerable room for improvement. This study presents two fractional order models and conducts parameter identification by integrating them with classical experimental data. The first model is an independent fractional order model. In comparison with traditional models, it demonstrates significantly higher fitting accuracy in frequency regions beyond the ultra - low frequency range. The second model is a segmented fractional order model, which determines segments according to critical frequencies. Although this model enhances the overall fitting accuracy, it also leads to increased complexity. To tackle this complexity issue, a rough design strategy is proposed to minimize the critical frequency. Research indicates that under such a strategy, the inertial effect dominates the behavior of the fluid inerter. Even when the inde-pendent fractional order model is used, a high fitting accuracy can be achieved. Consequently, by designing the structural parameters and fluid medium of the fluid inerter based on the rough design strategy, the model can be simplified. Moreover, compared with traditional nonlinear inerter models, the transfer function and eigenvalue analysis methods can be effectively applied. This enables the acquisition of more comprehensive theoretical research results, thereby greatly facilitating theoretical analysis.” |
Comments 2: The classical mathematical model (equations 5-6) should be described in more detail. Not all notations are given. The physical meaning of each term in the expression for the force F should be explained in the diagram. |
Response 2: Agree. We have added specific meanings of some variables in equations 5-6, such as the following in red font: “?1 is the effective area of the hydraulic cylinder, ?2 is the section area of the helical channel, l is the unfolded length of the helical tube, and mhel is the equivalent mass of the fluid inside the helical tube and L is the height of the helical tube component. The other variables in the equivalent coefficient are the structural parameters of the inerter and the fluid parameters, as shown in Table 1.”
|
Comments 3: The work uses two expressions for the force, for laminar (5) and for turbulent flow (equation 6). How do the authors choose which one to choose?
|
Response 3: Thank you for pointing this out. We provide an explanation on how to choose equations 5 and 6, as shown in red font below: “Equations (5) and (6) are derived using corresponding classical theories based on the assumption that the medium flow inside the helical tube is dominated by one of them. However, in actual flow conditions, it is generally difficult to accurately determine which one is dominant, so nonlinear correction is still necessary. The author's fractional order model does not need to discuss which one is dominant, only needs to fit the experimental data to obtain the fractional derivative order. The derivative order can reflect the approximate proportion of inertia and damping, such as damping being dominant between 0.5-1.5 and inertia being dominant between 1.5-2.”
|
Comments 4: The main problem with the work is that the use of a model with fractional dimension is not justified. It remains unclear to the end why the authors needed to use a model with fractional dimension. What physical meaning does it carry? The authors' assertion that it better describes the experiment is not convincing. In my opinion, this is just a question of approximation. We are considering a classical problem and it should be well described by the equations of classical mechanics. With the same success, using the least squares method, the parameters of the classical model can be fitted in the best possible way, and the classical model will describe the experiment better than the model with fractional dimension.
|
Response 4: Thank you for pointing this out. We have provided the following explanation for the rationality of fractional order models: (1) Firstly, for the complex flow patterns of fluids in helical tubes, it is not possible to accurately represent them with a single classical model, and high-order polynomials are usually added to improve fitting accuracy; (2) high-order polynomials introduce nonlinear terms, which increase the difficulty of theoretical research. For example, introducing nonlinear terms and small changes in initial values may lead to unexpected phenomena; (3) fractional order models have fewer parameters, such as derivative orders 1-2 representing fluid solid interaction characteristics and 0-1 representing viscoelastic characteristics; (4) the design goal of fluid inerters is to focus on inertia, that is, the working mode dominated by turbulence. One explanation for turbulence is chaos, which is mathematically considered a fractional dimensional system, and the dynamic characteristics of fractional dimensional systems can be described using fractional order systems [14]. |

Reviewer 3 Report
Comments and Suggestions for Authors
The submitted manuscript introduced a fractional-order dynamic model for a fluid-based inerter, which supplement previous literatures in this space with more in-depth discussion with different frequency regions and physical meanings of different parameters and their values. I found the derivation process of equations to be thorough and sound. The model was validated with previous results in literature, which showed a favorable comparison and improved accuracy with the new model. However, please provide more descriptions to the data sets in text.
Comments on the Quality of English LanguageThe English needs to be improved for this manuscript as I often find myself guess what authors mean. It can be benefited with editing works from a native speaker. The in-text reference must be improved and corrected to the correct style.
Author Response
Comments 1: The submitted manuscript introduced a fractional-order dynamic model for a fluid-based inerter, which supplement previous literatures in this space with more in-depth discussion with different frequency regions and physical meanings of different parameters and their values. I found the derivation process of equations to be thorough and sound. The model was validated with previous results in literature, which showed a favorable comparison and improved accuracy with the new model. However, please provide more descriptions to the data sets in text.
|
|
Response 1: Thank you for pointing this out. Regarding the above comment, we provide the following explanation: Firstly, the applicability of this model is also the focus of this article to verify. 1) Selected experimental data from two fluid inertial containers with completely different structural dimensions proposed by two scholars [12] [10]; 2) Extended to other frequencies outside the experiment (as shown in Figures 3 and 5), and compared with traditional models for numerical verification, the results showed that fractional order models can achieve better fitting accuracy. 3) If possible, in the future, according to the design rules in this article, multiple prototypes will be designed to measure data at more frequencies to verify the universality of the model. The data set provided in the paper is mainly divided into two parts: (1) the dataset obtained from actual experimental curves (as shown in Figure 2 and Figure 4). The text description states: "The Coulomb friction force amplitudes in references [12] and [10] are approximately 400N and 550N, respectively, and the equivalent inertance b is 370.38 and 73.82, respectively. The experimental data of the first and second groups (Figure 6 in reference [12]) were obtained under the action of sin displacement functions with Ω=0.5Hz, A0=20mm and Ω=12 Hz, A0 = 5mm, respectively, the third set of experimental data (Figure 4 in reference [10]) was obtained under the action of a sine displacement function with Ω = 3Hz and A0 = 17.5mm.” (2) In addition to the known experiments in references [12] and [10], datasets with different frequencies and amplitudes were obtained based on traditional and fractional order models (as shown in Figures 3 and 5). 2. Response to Comments on the Quality of English Language Point 1: The English needs to be improved for this manuscript as I often find myself guess what authors mean. It can be benefited with editing works from a native speaker. The in-text reference must be improved and corrected to the correct style. Response 1: Regarding English grammar and expression issues, the entire article has been revised and marked in red in the paper. |
|

Reviewer 4 Report
Comments and Suggestions for Authors
Please see the attached file.

Author Response
Comments 1: How generalizable is the proposed fractional order model across other fluid inerter configurations or operating conditions? Can the authors demonstrate that their model is not overfitting to a narrow data regime? |
|
Response 1: Thank you for pointing this out. Regarding the above comment, we provide the following explanation: “Firstly, the applicability of this model is also the focus of this article to verify. 1) Selected experimental data from two fluid inertial containers with completely different structural dimensions proposed by two scholars [12] [10]; 2) Extended to other frequencies outside the experiment (as shown in Figures 3 and 5), and compared with traditional models for numerical verification, the results showed that fractional order models can achieve better fitting accuracy. 3) If possible, in the future, according to the design rules in this article, multiple prototypes will be designed to measure data at more frequencies to verify the universality of the model.” |
|
Comments 2: Can the authors provide experimental or numerical validation of the rough design strategy proposed in Equation (16)? |
|
Response 2: Thank you for pointing this out. Regarding the above comment, we provide the following explanation: “(1) Equation (16) is obtained by substituting the structural parameters of the inertial container into Equation (15), and Figure 7 is plotted based on Equations (13) and (15). The results in the figure have demonstrated that inertial containers with different structural parameters can have different critical frequencies. (2) In addition, reducing to formula 16 can better illustrate which structural parameters are related, and the essence of the two formulas is the same, so no further numerical simulation was conducted.” |
|
Comments 3: While the manuscript introduces fractional order parameters such as ? and ?0, the physical interpretation and range of applicability for these parameters could be more clearly linked to the mechanics of fluid inerters. Providing dimensionless forms or normalized interpretations would enhance understanding. |
|
Response 3: Thank you for pointing this out. We provide an explanation on how to choose equations 5 and 6, as shown in red font below: “The dimension of the fractional derivative part in Equation (7) is fractional order, and the coefficient order in front of it is also fractional order. Only when two terms are multiplied can there be a clear dimension, so even normalization cannot avoid the existence of dimensions. The particularity of fractional derivative terms: 1) A term can simultaneously express two characteristics, and which category it belongs to is mainly determined by the order of the fractional derivative, while the preceding coefficients determine the overall size; 2) When the order is an integer, it can degenerate into the traditional definition of a specific class of characteristics. This article proposes a modified approach based on traditional parameters (inertance b) to reduce parameter and fitting costs.” |
|
Comments 4: The model relies on parameters identified via a genetic algorithm. Sensitivity analysis for ?, ?0, and ?0 would give insight into the robustness and reliability of the fitted models. |
|
Response 4: Thank you for pointing this out. The model identification method is not limited to genetic algorithms, and the selection of parameter ranges does have sensitivity issues, and it should be as close as possible to the actual physical category. We have provided a textual description of the approximate selection range of model parameters in the article, as follows: “The model parameters ?, b0, and f0 identified by genetic algorithm (GA) have a certain impact on the robustness and reliability of the model fitting due to their selection range. This article provides the following explanation on the selection of three parameters: (1) The setting of ? is related to frequency selection. At ultra-low frequencies, damping is clearly the main factor, and ? is set to 0-1.5, which is more appropriate. At other frequencies, inertia is the main factor, and ? is set between 1.5-2; (2) b0 refers to the addition of b0 times b to the traditional inertance b, with a range of 0-30. (3) f0 is a constant Coulomb friction force determined by low-frequency experiments.” |
|
Comments 5: While references are comprehensive, the comparison to other modern fractional models is absent. Positioning this work relative to recent advancements would increase its relevance. Response 5: Thank you for pointing this out. In the previous research work [16, 17], the author team partially verified the rationality of using fractional order models for inerters from a mathematical perspective, while this article further verifies its effectiveness from a physical perspective. In addition, fractional order models have been successfully applied in various multiphase mechanical properties [14, 15, 20], as briefly introduced in the introduction. Comments 6: The conclusion currently repeats earlier claims without summarizing broader implications or future directions. So, highlight the novelty, practical implications, and suggest future extensions. Response 6: Thank you for pointing this out. We have made comprehensive improvements to the conclusion section based on your suggestions, as follows: “This article innovatively proposes the concept of critical frequency using classical theories and applies it to the design of fluid inerters with inertial characteristics as the focus. At the same time, fully utilizing the advantages of fractional order models in fitting multiphase characteristics, two fractional order models with clear physical meanings and more simplified parameters were constructed based on traditional fluid in-erter models. The specific research results are as follows: (1) Research has shown that when using segmented fractional order models for fluid inerters, the fitting accuracy in the ultra-low frequency region is better than that of independent fractional order models. However, this high precision comes at the cost of increasing model complexity. This suggests that we need to balance the relationship between accuracy and model complexity in practical applications. (2) Research has found that when the critical frequency is small enough, the use of an independent fractional order model for fluid inerters can meet practical engineering needs. Equation (16) can serve as a rough design principle for fluid inertial containers, providing a simple and effective reference for engineering design. Overall, by designing the structural parameters of fluid inertia and fluid medium reasonably, the fractional order model demonstrates good applicability. Compared with traditional nonlinear inertial models, fractional order models can use transfer function and eigenvalue analysis methods to conduct theoretical research, which can obtain richer theoretical results and greatly facilitate theoretical analysis work. This not only deepens the theoretical understanding of fluid inertial containers but also provides strong support for the promotion and application of fluid inertial containers. Overall, by designing the structural parameters and fluid medium of the fluid inerter reasonably, the fractional order model demonstrates good applicability. Com-pared with traditional nonlinear inerter models, fractional order models can use trans-fer function and eigenvalue analysis methods to conduct theoretical research, which can obtain richer theoretical results and greatly facilitate theoretical analysis work. This not only deepens the theoretical understanding of fluid inerters but also provides strong support for the promotion and application of fluid inerters.”
|
|

Round 2
Reviewer 2 Report
Comments and Suggestions for Authors
The authors answered all my questions.
Now, in my opinion, the article can be published.
Author Response
Dear Reviewers, We sincerely thank you and the Reviewers for your careful evaluation and constructive comments. All suggestions have been thoroughly addressed and incorporated into the revised manuscript. The modifications, highlighted in red (or tracked changes). We deeply appreciate the Reviewers' expertise and look forward to the prompt publication of this work. |
||
|
|
|